# Profiling the medical, functional, cognitive, and psychosocial care needs of adults assessed for home care in Ontario, Canada: The case for long-term 'life care' at home

**Margaret E. Saari** [1,2]*, **Justine L. Giosa**[1,3], **Paul Holyoke**[1], **George A. Heckman**[3], **John P. Hirdes**[3]

1 SE Research Centre, SE Health, Markham, Ontario, Canada, 2 Lawrence S. Bloomberg Faculty of Nursing, University of Toronto, Toronto, Ontario, Canada, 3 School of Public Health Sciences, University of Waterloo, Waterloo, Ontario, Canada

* margaretsaari@sehc.com

**Data Availability Statement:** RAI-Home Care data utilized for this study is housed within the Home

## Abstract

Calls to leverage routinely collected data to inform health system improvements have been made. Misalignment between home care services and client needs can result in poor client, caregiver, and system outcomes. To inform development of an integrated model of community-based home care, grounded in a holistic definition of health, comprehensive clinical profiles were created using Ontario, Canada home care assessment data. Retrospective, cross-sectional analyses of 2017–2018 Resident Assessment Instrument Home Care (RAI-HC) assessments (n = 162,523) were completed to group home care clients by service needs and generate comprehensive profiles of each group's dominant medical, functional, cognitive, and psychosocial care needs. Six unique groups were identified, with care profiles representing home care clients living with Geriatric Syndromes, Medical Complexity, Cognitive Impairment and Behaviours, Caregiver Distress and Social Frailty. Depending on group membership, between 51% and 81% of clients had identified care needs spanning four or more Positive Health dimensions, demonstrating both the heterogeneity and complexity of clients served by home care. Comprehensive clinical profiles, developed from routinely collected assessment data, support a future-focused, evidence-informed, and community-engaged approach to research and practice in integrated home-based health and social care.

## Introduction

Aging Canadians want care choices, often preferring to live and age at home [1]. Demand for community-based care is predicted to significantly increase as population growth occurs in the oldest age stratum, culminating in over 2.7 million Canadians over age 85 by 2050 [2]. Within Ontario, Canada's most populous province, demand for community-based care is predicted to increase by 120 percent [2].

Care Database, owned by the Canadian Institute for Health Information and provided in de-identified, encrypted form to the University of Waterloo. No other sites are permitted to receive these data under this data sharing agreement. Access to the Home Care Database may be sought upon reasonable and justifiable request from the Canadian Institute for Health Information. Data requests may be made using CIHI's Data Request Form: https://www.cihi.ca/en/access-data-and-reports/make-a-data-request.

**Funding:** This study was funded by an unrestricted grant provided by SE Health to the SE Research Centre as part of the organization's commitment to impact-oriented health services research as a recognized research institution, learning health system and social enterprise. MS, JLG, and PH receive unrestricted salary support through their roles in the SE Research Centre. GAH receives unrestricted salary support from the Schlegel Chair in Geriatric Medicine at the Schlegel-UW Research Institute for Aging. The funders had no role in study design, data collection and analysis, decision to publish, or preparation of the manuscript.

**Competing interests:** The authors have declared that no competing interests exist.

To remain in their community, a person's combination of self-care abilities, family support and formal care must match their level of need. A range of supports may be required including assistance with personal care, household chores, meals, social engagement, medical care, chronic disease management, and transportation [3–5]. Home and community care services are intended to assist individuals with these types of supports; however, services are not insured under the Canada Health Act and vary significantly by jurisdiction [6, 7]. Differences in funding and eligibility criteria, as well as service availability, result in service constraints and the prioritization of episodic "illness care", with most supportive services focusing on activities of daily living [7].

Lack of alignment between service offerings and population health and social care needs results in dependence on caregivers to meet needs, or they simply go unmet [8, 9]. Unmet needs result in poor outcomes such as reduced quality of life [7, 10, 11], premature admission to facility-based care [8], emergency department visits [12], hospitalizations [13] and even death [14]. The extent to which home care needs are accurately identified and met is therefore not only relevant to the individual and their caregiver's well-being, but also supports more efficient use of scarce resources across the healthcare system.

Most literature examining home care client needs focuses on a single need or domain, often functional impairment [3, 11–14]. Recent studies have found that between 23% and 54% of home care clients have unmet functional and supportive care needs [7, 10, 12]. Broadening the scope of care to include needs that maintain independence, such as access to assistive technologies, home modifications, and transportation, increases the prevalence of unmet needs in home care to 80% [13].

While functional impairment is an important factor in admission to facility-based long-term care (LTC), the pathways to admission are not always straightforward. Social frailty [15–17], caregiver distress [18, 19], chronic disease management needs [20], cognitive impairment and expressive behaviours [15, 21, 22], medical complexity [23] and geriatric syndromes [16] have all been linked to the need for facility based LTC. These factors represent care needs spanning medical, functional, cognitive, and psychosocial domains, pointing towards the need for a broader conceptualization of health and subsequently, health care. Researchers in the Netherlands proposed an expanded definition of health, called Positive Health, which includes dimensions typically associated with "health" such as daily functioning and bodily functions, but also encompasses aspects of societal participation, quality of life, meaningfulness, and mental well-being [24, 25]. We propose the term 'life care' to describe the complementary health and social care required to meet the holistic care needs included in Positive Health. To inform health system planning required to adopt a broadened definition of health and expand service offerings to provide 'life care', an understanding of the prevalence of care needs reflecting the Positive Health dimensions is required.

Critical to health system planning and evaluation is a source of high-quality, reliable data [26]. Routinely collected standardized health assessments can provide these data, with their longitudinal nature meeting calls for the utilization of common data elements and core outcome measures to understand populations and assess the outcome(s) of care in a way that supports evidence-based decision-making [27, 28]. Across Ontario, more than 150,000 home care clients are assessed each year with a standardized, comprehensive home care assessment [29, 30]. These data are collected as part of routine clinical processes and support systematic evaluation of home care client needs, ranging from symptomology (e.g., pain, dyspnea) to functional independence (e.g., mobility with assistive devices) to social support (e.g., presence and level of caregiver support). These data have been previously used to evaluate quality of existing care models [31, 32], identify opportunities for system integration and improvements [33, 34] and support identification of populations requiring targeted interventions [19, 35, 36].

Given the breadth of information collected in these assessments, their documented reliability and validity and their widespread implementation, these assessments provide an ideal data source for understanding dominant 'life care' needs at a population level. In a recent study, information included in this home care assessment was found to map to all six dimensions of Positive Health, making it an ideal tool to examine 'life care' needs [37]. To date, most of the research using these data have focused on understanding the clinical profiles of sub-groups of home care clients based on diagnosis [38–40] or service type [41]. However, home care services are not typically organized or delivered by diagnostic sub-group, so these studies have limited application in development of a new model of integrated home care. Grouping clients based on the types and intensity of their identified care needs, rather than by diagnosis, location or even care approach, allows for the development of home care service models which are driven by the influence of care needs on health and functioning, regardless of their etiology [42]. This approach takes into account the complexity added to the care situation through the convergence of medical, functional, social and environmental factors [43].

## Objective

The primary research question for this study is: *What are the dominant medical, functional, cognitive, and psychosocial 'life care' needs of distinct groups of community-dwelling adults assessed for Ontario home care services?* It was hypothesized that the groups would have varying dominant care needs and that these dominant care needs would align with known predictors of admission to LTC facilities. This work contributes to a larger mixed-methods study aiming to develop a new LTC model which meets 'life care' needs of aging Ontarians in their homes, thereby expanding care options [44].

## Methods

### Study design

In this paper, we report on the retrospective, cross-sectional analyses of routinely collected health data conducted in Phase 1 of the larger mixed-methods study, specifically focused on understanding the 'life care' needs of community-dwelling adults assessed for long-stay home care services in Ontario, Canada. These quantitative analyses will serve as the foundation for engagement of a wide range of experts-by-experience in the design of a new model of community-based long-term care. Detailed methods of the larger mixed-methods study have been reported elsewhere [44]. Reporting of this study follows the RECORD (REporting of studies Conducted using Observational Routinely-collected health Data) reporting guidelines [45].

### Setting

Ontario home care services are managed by 14 Home and Community Care Support Services (HCCSS) organizations. Clients referred for service(s) undergo initial screening assessment to determine eligibility, and an essential services plan is established. A comprehensive assessment is completed with clients expected to receive long-stay services (60 days or longer) [46]. Regulated health professionals in care coordination roles with HCCSS complete these assessments and determine eligibility for nursing, physiotherapy, occupational therapy, speech language therapy, social work, nutrition services, personal care, and homemaking services as well as home healthcare supplies [47]. Contracted service provider agencies then deliver direct patient care. Responsibility for on-going case management, placement services for facility based LTC and referrals to community support services remains with care coordinators.

## Data

Study data were drawn from the Resident Assessment Instrument-Home Care (RAI-HC), a standardized comprehensive assessment employed in 10 Canadian provinces and territories to assess care needs and allocate services for long-stay home care clients. This tool is developed and maintained by interRAI, an international, not-for-profit, network of researchers and practitioners from over 35 countries committed to evidence-informed clinical practice and decision-making [48]. The interRAI Home Care assessment system consists of an assessment form, health index scales used for outcome measurement, clinical assessment protocols (CAPs) supporting individualized care planning and a case mix index [30]. The assessment form contains over 300 items organized into 20 sections (e.g., communication and vision, functional status, skin condition, social supports etc.). In Canada, assessment data are submitted to the Canadian Institutes of Health Information (CIHI), who maintain the national data repository. CIHI uses several measures to ensure data quality and completeness through the submission process. Most individual data elements in the RAI-HC are mandatory, including all elements used to derive the key outputs including health index scales and CAPs [49]. Assessment data were accessed on April 2$^{nd}$, 2020, through a licensing agreement between University of Waterloo and CIHI. Use of data for secondary analysis and the processes in place to protect patient and confidentiality received ethics clearance from the Office of Research Ethics at the University of Waterloo (ORE#30173).

## Sample

Analyses were completed on the population of Ontario home care clients assessed in the 2017/2018 fiscal year (April 1$^{st}$, 2017, to March 31$^{st}$, 2018). Client records were included for persons: 1) 20 years and older at time of assessment; 2) assessed with the RAI-HC for long-stay home care or as part of the facility-based LTC referral process; and 3) assessed in a hospital or community care setting. Hospital-based assessments were included as they are used to inform hospital discharge planning, including home care service plans [50]. For persons with multiple assessments within the observation window, the most recent assessment within the fiscal year was included (n = 205,405). Note, not all clients assessed for services are deemed eligible for and/or choose to receive care.

## Analysis

A three-step segmentation process identified distinct groups of home care clients comprised of individuals whose characteristics place them at risk for facility-based LTC admission. First, the population was grouped into six Service Levels using an existing hierarchical algorithm originally developed to identify individuals requiring comprehensive assessment and care coordination services during the COVID-19 pandemic [51]. To assign clients to a unique group, each client is first assessed against inclusion criteria for Service Level 1 and assigned to this group if criteria are met. If criteria are not met, they are assessed against inclusion criteria for Service Level 2. Again, if criteria are met, they are assigned to this group but if criteria are not met, they are assessed against inclusion criteria for Service Level three and so on, until they reach Service Level 6. The presence of geriatric syndromes, medical instability, functional impairments, risk of long-term care placement, caregiver distress and cognitive ability are considered in the algorithm, with each subsequent Service Level requiring less intensive service allocation and care coordination [51]. Inclusion criteria and proposed system level service requirements for each group are available in **Table 1**. A plain language summary of the coding rules for the service levels algorithm are included as **S1 Table**.

**Table 1. System focus and inclusion criteria for Service Level grouping algorithm.**

| | Service Level 1 | Service Level 2 | Service Level 3 | Service Level 4 | Service Level 5 | Service Level 6 |
|---|---|---|---|---|---|---|
| **System Focus** | Care coordination with Specialized Geriatric Services Involvement | Care Coordination with Primary Care Monitoring | Care Coordination with Long Term Home and Community Care, Complex Needs | Care Coordination with Long Term Home and Community Care, Moderate Needs | Care Coordination with Chronic Disease Management Support | On-going Monitoring with Self-Report Assessment |
| **Group Inclusion Criteria** | 1. Four or more of the Geriatric 5Ms with high or very high service needs and high medical instability | 1. High medical instability and high or very high service needs 2. High medical instability and elevated risk of death related to COVID-19 3. Highest medial instability 4. Highest risk of death related to COVID-19 | 1. Lower medical instability and high service needs 2. Lower medical instability and very high service needs 3. Caregiver distress with caregiver providing more than 14h care per week | 1. Cognitive impairment and moderate service needs 2. Cognitive impairment and high medical instability | 1. Intact cognition with lower medical instability and moderate service needs | 1. Intact cognition with lower medical instability and low service needs |

In the second step, the six Service Levels were reviewed by the research team. Using a broad conceptualization of known predictors of facility-based LTC admission including social frailty [15–17], caregiver distress [18, 19], chronic disease management needs [20], cognitive impairment and expressive behaviours [15, 21, 22], medical complexity [23] and geriatric syndromes [16, 21] and considering feasibility for subsequent phases of the larger research study, it was determined that only Service Levels 1 through 4 would be included for deeper analysis of their 'life care' needs. Given the high proportion of unmet needs of home care clients, we chose to pragmatically scope the project to target a population with more complex needs at higher risk of facility-based long-term care admission based on well-established risk algorithms [52–54]. Based on the grouping algorithm, individuals included in Service Levels 5 and 6 are cognitively intact with some medical instability and low to moderate care needs and therefore, not within the target population. Additionally, as subsequent phases of the planned mixed method study are quite resource intensive, this was also a feasibility decision. Given the size (n = 81,699) of Service Level 3, and the variability in care needs represented by the inclusion criteria, this group was sub-divided using the three sets of inclusion criteria (**Table 2**). This process resulted in a total sample of 162,523 divided into six unique groups for analysis.

To describe the 'life care' needs of clients in these six groups, descriptive analyses were conducted by the lead author using Statistical Analysis System (SAS) Version 9.4. Demographic and clinical variables are presented as counts and percentages for categorical variables and means and standard deviations for continuous variables. To determine statistical significance of differences in care needs across groups, we performed a series of chi-square tests. Given the large number of comparisons made, a Bonferroni adjustment [55] was made such that the adjusted alpha was 0.0015 (0.05/33 tests). Finally, to support understanding of the relative intensity of care needs across groups and how care needs cluster within groups, we generated a heat map. Heat maps are ideal for exploratory analyses of large datasets with complex multivariate data [56]. In a heat map, each matrix entry is represented by a colour that corresponds to its magnitude. In this analysis we chose to analyze the frequency distribution of each care need individually to draw attention to rarer care needs which have important planning and/or operational considerations, such as education and training, identifying skill or provider mix or visit frequency, in a new model of integrated home and community care. For the heat map, a quantile colour mapping approach was selected, using percentiles to determine where colour transitions take place in the heat map [56]. A divergent colour gradient, ranging from blue

**Table 2. Final analysis groups following segmentation procedure.**

| | Service Level 1 | Service Level 2 | Service Level 3 | | | Service Level 4 |
|---|---|---|---|---|---|---|
| **System Focus** | Care coordination with Specialized Geriatric Services Involvement | Care Coordination with Primary Care Monitoring | Care Coordination with Long Term Home and Community Care, Complex Needs | | | Care Coordination with Long Term Home and Community Care, Moderate Needs |
| **Analysis Group** | Group A | Group B | Group C | Group D | Group E | Group F |
| **Group Inclusion Criteria** | 1. Four or more of the Geriatric 5Ms with high or very high service needs and high medical instability | 1. High medical instability and high or very high service needs 2. High medical instability and elevated risk of death related to COVID-19 3. Highest medial instability 4. Highest risk of death related to COVID-19 | 1. Lower medical instability and very high service needs | 1. Lower medical instability and high service needs | 1. Caregiver distress with caregiver providing more than 14h care per week | 1. Cognitive impairment and moderate service needs 2. Cognitive impairment and high medical instability |

($10^{th}$ percentile) to white ($50^{th}$ percentile) to red ($90^{th}$ percentile), was used to visualize the relative intensity of each care need across the six groups.

## Measurement approach

Demographic variables were drawn directly from RAI-HC items. To protect client confidentiality, a rurality measure was generated by CIHI prior to data transfer, using client postal codes [57, 58]. Clinical variables measuring home care client life care needs included assessment outputs (e.g., health index scales, clinical assessment protocols) as well as individual assessment items. Health index scales focus on clinical, functional, cognitive, or social issues to support risk assessment and outcome measurement. CAPs use embedded algorithms to bring together items from across the assessment to highlight priority issues which may require further investigation and intervention [59, 60]. **Table 3** provides a summary of the measurement approach used to examine life care needs using RAI-HC data sources (health index scales, CAPs, and individual items), including the life care needs examined, and a description of included clients. Variables were selected and grouped by Positive Health dimensions based on Fowokan et al.'s mapping work [37].

## Findings

### Sample description

Following segmentation, 162,523 home care clients, representing 79% of the total population assessed were assigned to the six client groups. Groups ranged in size, containing between 6% and 31% of clients assessed. A description of the demographic characteristics of the full study sample is provided first, followed by in-depth examination of the demographics and life care needs of the six unique client groups.

Participants in the full study sample ranged in age from 20 to 112 years old, with 85 years and older being the largest (43.0%, n = 69,882) age group. Most (61.4%, n = 99,702) home care clients were female and 38.3% (n = 61,160) were married. The greatest proportion of clients

**Table 3. Measurement approach to examine life care needs using the RAI-HC, grouped by Positive Health dimensions.**

| Life Care Need | RAI-HC Data Source(s) | Description of included clients |
|---|---|---|
| **Bodily Functions** | | |
| Medical instability | Changes in Health, End-stage disease, and Signs and Symptoms (CHESS) scale [52] | Persons with a score of 2 or more on the CHESS scale, indicating at least low health instability. |
| Pain | Pain CAP [59] | Persons experiencing daily pain or those experiencing pain they describe as "severe, horrible or excruciating" whether it occurs daily or less frequently. |
| Cardiorespiratory symptoms | Cardiorespiratory Conditions CAP [59] | Persons experiencing chest pain, shortness of breath, irregular pulse, or dizziness. |
| Wound care | Pressure Ulcer CAP [59] Section L Skin Condition items [30] | Persons with a current pressure ulcer, stasis ulcer, burn, lesion, skin tear or surgical wound. |
| Catheter care | Section H Continence items [30] | Persons utilizing a catheter (indwelling or intermittent) to support urinary continence. |
| Ostomy care | Section H Continence items [30] Section N Treatments and Procedures items [30] | Persons with an ostomy to support bowel continence or those with a tracheostomy. |
| Dehydration | Dehydration CAP [59] | Persons assessed as being dehydrated and/or receiving insufficient fluids with at least one identified symptom of dehydration. |
| Nutritional concerns | Section K Oral and Nutritional Status items [30] | Persons with unintended weight loss (5% in 30 days or 10% in 180 days) or consistently consuming one or fewer meals. |
| Medication concerns | Appropriate Medications CAP [59] | Persons receiving 9 or more medications and experiencing potential adverse effects such as chest pain, dizziness, edema, shortness of breath, poor health, or recent deterioration |
| Other medical interventions and treatments | Section N Treatments and Procedures items [30] | Persons with scheduled therapies or programs requiring care or monitoring including chemotherapy, IV therapy, medication by injection, radiation treatments, dialysis etc. |
| **Daily Functioning** | | |
| Assistance with Instrumental Activities of Daily Living (IADL) | IADL Capacity Hierarchy Scale [61] | Persons who are dependent in any of the following IADLs: meal preparation, housework, managing finances, managing medications or shopping. |
| Assistance with Activities of Daily Living (ADL)–Supervision or guidance | ADL Hierarchy Scale [61, 62] | Persons who require supervision or guidance to complete ADLs including personal hygiene, locomotion, toileting or eating. |
| Assistance with ADLs–Physical assistance | ADL Hierarchy Scale [61, 62] | Persons who require any physical assistance to complete ADLs including personal hygiene, locomotion, toileting or eating. |
| Continence care | Section H Continence item [30] | Persons experiencing frequent incontinence episodes (twice weekly or more). |
| Falls | Falls CAP [59] | Persons experiencing one or more falls in the past 90 days. |
| Unsteady gait | Section J Health Condition item [30] | Persons identified as having an unsteady gait |
| Low levels of physical activity | Physical Activities Promotion CAP [59] | Persons identified as having low levels of physical activity and indicators of functional reserves to support increased physical activity |
| **Mental Wellbeing** | | |
| Cognitive impairment—Mild | Cognitive Performance Scale [54] | Persons with mild cognitive impairment |
| Cognitive impairment–Moderate to severe | Cognitive Performance Scale [54] | Persons with moderate to severe cognitive impairment |
| Communication | Communication CAP [59] | Persons identified as having deficits in expressive or receptive communication |
| Delirium | Delirium CAP [59] | Persons with active symptoms of delirium |
| Behaviours | Section E Mood and Behaviours items [30] | Persons exhibiting any behaviours including wandering, verbal or physical abuse, socially inappropriate behaviour or resisting care. |
| Mood concerns | Depression Rating Scale [63] | Persons exhibiting 3 or more indicators of mood disturbance. |
| **Quality of Life** | | |
| Loneliness | Section F Psychosocial Well-Being item [30] | Persons who say or indicate they are lonely. |
| Home environment concerns | Section O Environmental Assessment items [30] | Persons who live in a home environment where there are physical safety concerns. |
| Risk of abuse or neglect | Abusive relationship CAP [59] | Persons with potential indicators of abuse including being fearful of a family member or caregiver, unusually poor hygiene, or have been observed to be neglected, abused, or mistreated. |

*(Continued)*

**Table 3.** (Continued)

| Life Care Need | RAI-HC Data Source(s) | Description of included clients |
|---|---|---|
| **Participation** | | |
| Social participation | Section F Psychosocial Well-Being item [30] | Persons who report feeling distressed due to a decline in their social activities. |
| Informal support concerns | Section P Social Support item [30] | Persons without a caregiver or whose caregiver reports being unable to continue. |
| Caregiver distress | Section P Social Support item [30] | Persons with a primary caregiver who reports feelings of distress, anger, or depression. |

(44.1%, n = 71,632) did not co-reside with their primary caregiver and primary caregivers were most likely to be a child or child -in-law (49.6%, n = 80,553). About 1 in 10 clients (13.4%, n = 21,721) lived in rurally and 1 in 13 (7.2%, n = 11,706) were assessed for services in hospital.

**Table 4** summarizes the demographic characteristics of each of the six client groups. Demographically, Groups A through D are similar: older in age, predominantly female with a co-resident caregiver. In these groups, the main differentiator lies in common disease diagnoses and number of comorbidities observed. Groups A and B have the highest prevalence of chronic conditions such as arthritis, heart failure and diabetes, and the most comorbidities. Those in Group B are twice as likely to have renal failure than Group A and more than 10 times as likely than other groups. In Group C, more than 60% of clients have dementia. No single diagnosis stands out in Group D. However, more striking are demographic differences seen in Group E and F. Group E is the youngest, with a high proportion of male clients who are married and depend on their co-resident spouse as their primary caregiver. Group F has the highest **proportion** of clients over the age of 85 who are predominantly female and living alone. Most clients in Group F identified a child as their primary caregiver; however, this group also has the highest proportion (17.6%) of primary caregivers in the "other" category which includes friends, neighbours, siblings etc.

**Table 5** presents the frequency of client 'life care' needs in each group, categorized by Positive Health dimensions, and ordered from highest frequency care need to lowest. Statistically significant differences in care need frequency were found for all care needs examined. **Table 6** presents a heat map visualizing client 'life care' needs across the six groups, highlighting the variation in care needs between groups. The divergent colour gradient, ranging from blue (10th percentile) to white (50th percentile) to red (90th percentile), shows the relative intensity of each care need across the groups and allows for visualization of how care needs cluster within each of the six unique groups.

Several 'life care' needs were found to be common to all care groups, highlighting important clinical issues across the home care population. As expected, care needs related to daily functioning were prevalent, with most clients being dependent in at least one instrumental activity of daily living (IADL), at least 50% of each group requiring supervision or assistance with activities of daily living (ADL) and the majority having an unsteady gait. Given minimal variation and high prevalence of IADL care needs, a more in-depth examination of IADL dependencies across the groups is provided (**Fig 1**). Dependencies in physical IADLs, such as housework, are consistent across groups. However, greater variation is seen in cognitive IADLs, such as managing medications and phone use. Other common care needs include experiencing daily or excruciating pain (44% to 69%), as well as living with incontinence (42% to 63%).

**Table 4. Demographic characteristics by client group.**

| | Group A | Group B | Group C | Group D | Group E | Group F |
|---|---|---|---|---|---|---|
| | 19.3% | 18.9% | 13.3% | 31.0% | 6.0% | 11.6% |
| | (n = 31,284) | (n = 30,793) | (n = 21,611) | (n = 50,314) | (n = 9,674) | (n = 18,847) |
| **Age group, n (%)** | | | | | | |
| <65 y | 2,730 | 2,979 | 2,883 | 7,788 | 1,641 | 1,915 |
| | (8.7%) | (9.7%) | (13.3%) | (15.5%) | (17.0%) | (10.2%) |
| 65–74 y | 4,236 | 4,975 | 2,787 | 7,243 | 1,662 | 2,323 |
| | (13.5%) | (16.2%) | (12.9%) | (14.4%) | (17.2%) | (12.3%) |
| 75–84 y | 9,868 | 10,042 | 6,662 | 14,637 | 3,015 | 5,255 |
| | (31.5%) | (32.6%) | (30.8%) | (29.1%) | (31.2%) | (27.9%) |
| 85+ y | 14,452 | 12,797 | 9,279 | 20,646 | 3,356 | 9,354 |
| | (46.2%) | (41.6%) | (42.9%) | (41.0%) | (34.7%) | (49.6%) |
| **Female, n** | 19,170 | 18,553 | 13,036 | 30,721 | 5,573 | 12,649 |
| (%) | (61.3%) | (60.3%) | (60.3%) | (61.1%) | (57.6%) | (67.1%) |
| **Married, n** | 11,641 | 11,991 | 8,054 | 18,176 | 5,787 | 5,511 |
| (%) | (37.8%) | (39.6%) | (37.8%) | (36.9%) | (60.6%) | (29.8%) |
| **No co-resident caregiver, n** | 14,635 | 13,457 | 9,395 | 22,699 | 1,266 | 10,180 |
| (%) | (46.8%) | (43.7%) | (43.5%) | (45.1%) | (13.1%) | (54.1%) |
| **Caregiver relationship** | | | | | | |
| Child, n | 16,748 | 15,565 | 10,145 | 23,850 | 3,606 | 10,642 |
| (%) | (53.5%) | (50.6%) | (46.9%) | (47.4%) | (37.2%) | (56.5%) |
| Spouse, n (%) | 9,100 | 9,607 | 6,511 | 14,217 | 4,995 | 3,927 |
| | (29.1%) | (31.2%) | (30.1%) | (28.3%) | (51.6%) | (20.8%) |
| Other, n | 4,210 | 4,384 | 4,005 | 9,655 | 1,004 | 3,315 |
| (%) | (13.5%) | (14.2%) | (18.5%) | (19.2%) | (10.4%) | (17.6%) |
| **Rural, n** | 4,651 | 4,758 | 2,684 | 6,377 | 1,122 | 2,129 |
| (%) | (14.9%) | (16.5%) | (13.2%) | (13.4%) | (12.2%) | (11.3%) |
| **Assessed in hospital, n** | 3,627 | 2,068 | 2,094 | 2,736 | 0 | 1,181 |
| (%) | (11.6%) | (6.7%) | (9.7%) | (5.4%) | (0%) | (6.3%) |
| **Number of comorbidities, mean** | 4.8 | 5.1 | 3.6 | 3.5 | 3.4 | 3.7 |
| (SD) | (2.2) | (2.4) | (1.8) | (1.8) | (1.8) | (1.8) |
| **Chronic disease diagnoses** | | | | | | |
| Arthritis, n | 18,405 | 17,789 | 9,374 | 24,107 | 4,872 | 10,302 |
| (%) | (58.8%) | (57.8%) | (43.4%) | (47.9%) | (50.4%) | (54.7%) |
| Heart disease, n | 9,846 | 13,106 | 3,625 | 9.646 | 1,815 | 3,756 |
| (%) | (31.5%) | (42.6%) | (16.8%) | (19.2%) | (18.8%) | (19.9%) |
| Dementia, n | 12,552 | 8,487 | 13,647 | 15,342 | 1,414 | 4,321 |
| (%) | (40.1%) | (27.6%) | (63.2%) | (30.5%) | (14.6%) | (22.9%) |
| Diabetes, n | 9,962 | 10,351 | 5,152 | 13,040 | 2,953 | 5,167 |
| (%) | (31.8%) | (33.6%) | (23.8%) | (25.9%) | (30.5%) | (27.4%) |
| Heart failure, n | 6,327 | 8,841 | 1,306 | 3,944 | 860 | 1,883 |
| (%) | (20.2%) | (28.7%) | (6.0%) | (7.8%) | (8.9%) | (10.0%) |
| Hypertension, n | 21,954 | 21,969 | 12,404 | 30,233 | 6,078 | 12,386 |
| (%) | (70.2%) | (71.3%) | (57.4%) | (60.1%) | (62.8%) | (65.7%) |
| Psychiatric diagnosis, n | 8,521 | 6,628 | 5,920 | 11,950 | 1,848 | 4,372 |
| (%) | (27.2%) | (21.5%) | (27.4%) | (23.8%) | (19.1%) | (23.2%) |
| Renal failure, n | 3,721 | 7,240 | 632 | 2,276 | 537 | 641 |

(*Continued*)

**Table 4.** (Continued)

| (%) | (11.9%) | (23.5%) | (2.9%) | (4.5%) | (5.6%) | (5.0%) |
|---|---|---|---|---|---|---|
| **Respiratory disease, n** | 8,471 | 12,860 | 1,884 | 6,099 | 1,122 | 2400 |
| (%) | (27.1%) | (41.8%) | (8.7%) | (12.1%) | (11.6%) | (12.7%) |
| **Stroke, n** | 6,633 | 8,112 | 3,665 | 9,121 | 1,724 | 3,384 |
| (%) | (21.2%) | (26.3%) | (17.0%) | (18.1%) | (17.8%) | (18.0%) |

## Dominant life care needs by group

Group A clients (n = 31,284) have complex needs, with the highest frequency of needs across all Positive Health dimensions, as well as most individual care needs examined. High needs related to Bodily Functions are observed in this group, with the highest rates of medical

**Table 5. Frequency of client life care needs, categorized by Positive Health dimensions.**

| | | Group A n = 31284 % | Group B n = 30793 % | Group C n = 21611 % | Group D n = 50314 % | Group E n = 9674 % | Group F n = 18847 % | Chi Square Test (p < 0.001) |
|---|---|---|---|---|---|---|---|---|
| Bodily Functions | Medical instability | 100.0 | 87.0 | 51.4 | 41.3 | 51.3 | 45.2 | ✓ |
| | Cardiorespiratory symptoms | 80.8 | 82.9 | 34.4 | 44.1 | 50.4 | 48.7 | ✓ |
| | Pain | 68.7 | 66.1 | 43.8 | 53.3 | 67.8 | 58.6 | ✓ |
| | Medication concerns | 62.9 | 51.6 | 18.5 | 23.9 | 33.5 | 28.8 | ✓ |
| | Nutritional concerns | 34.1 | 19.3 | 5.3 | 7.7 | 7.1 | 3.2 | ✓ |
| | Wound care | 27.8 | 25.1 | 15.5 | 16.9 | 21.3 | 16.3 | ✓ |
| | Other medical interventions and treatments | 26.3 | 33.9 | 12.5 | 14.9 | 18.5 | 15.4 | ✓ |
| | Dehydration | 9.0 | 3.6 | 0.9 | 0.8 | 1.1 | 0.6 | ✓ |
| | Catheter care | 6.0 | 5.7 | 3.4 | 4.1 | 7.6 | 5.5 | ✓ |
| | Ostomy care | 1.5 | 2.4 | 1.0 | 1.8 | 2.6 | 2.0 | ✓ |
| Daily Functioning | IADL dependence | 94.3 | 89.4 | 95.6 | 85.1 | 95.5 | 94.0 | ✓ |
| | Unsteady gait | 91.8 | 83.2 | 75.6 | 75.1 | 82.7 | 82.6 | ✓ |
| | Falls | 64.0 | 39.0 | 48.7 | 45.1 | 28.6 | 28.1 | ✓ |
| | Continence care | 59.9 | 47.8 | 62.6 | 42.0 | 43.1 | 49.0 | ✓ |
| | ADL physical assistance | 40.3 | 28.6 | 37.4 | 26.0 | 37.9 | 27.5 | ✓ |
| | ADL supervision/guidance | 34.1 | 43.0 | 32.2 | 27.1 | 49.4 | 72.5 | ✓ |
| | Low levels of physical activity | 25.0 | 18.7 | 11.7 | 10.8 | 18.5 | 11.1 | ✓ |
| Mental Wellbeing | Mild cognitive impairment | 51.4 | 51.6 | 40.8 | 59.4 | 56.4 | 100 | ✓ |
| | Mood concerns | 43.2 | 28.0 | 35.4 | 21.6 | 29.3 | 19.6 | ✓ |
| | Moderate / severe cognitive impairment | 38.0 | 12.6 | 56.8 | 28.4 | 0.0 | 0.0 | ✓ |
| | Communication | 34.8 | 26.2 | 37.3 | 33.1 | 23.9 | 37.4 | ✓ |
| | Behaviours | 25.5 | 8.0 | 68.6 | 2.9 | 0.6 | 0.7 | ✓ |
| | Delirium | 9.1 | 3.3 | 6.7 | 2.0 | 1.5 | 1.6 | ✓ |
| Quality of Life | Loneliness | 20.8 | 17.0 | 13.2 | 15.8 | 13.7 | 15.9 | ✓ |
| | Home environment safety concerns | 35.6 | 31.1 | 24.7 | 24.1 | 39.6 | 23.2 | ✓ |
| | Risk of abuse or neglect | 3.1 | 1.5 | 3.5 | 1.5 | 1.0 | 1.4 | ✓ |
| Participation | Caregiver distress | 57.2 | 37.8 | 55.3 | 33.8 | 85.0 | 18.2 | ✓ |
| | Informal support concerns | 37.8 | 25.6 | 35.5 | 23.7 | 48.5 | 16.3 | ✓ |
| | Social participation | 22.1 | 18.7 | 9.0 | 11.4 | 21.3 | 11.2 | ✓ |

**Table 6. Heat map to visualize client life care needs across groups, categorized by Positive Health dimension.**

| | | Group A | Group B | Group C | Group D | Group E | Group F |
|---|---|---|---|---|---|---|---|
| | | n = 31284 | n = 30793 | n = 21611 | n = 50314 | n = 9674 | n = 18847 |
| Bodily Functions | Medical instability | | | | | | |
| | Cardiorespiratory symptoms | | | | | | |
| | Pain | | | | | | |
| | Medication concerns | | | | | | |
| | Nutritional concerns | | | | | | |
| | Wound care | | | | | | |
| | Other medical interventions and treatments | | | | | | |
| | Dehydration | | | | | | |
| | Catheter care | | | | | | |
| | Ostomy care | | | | | | |
| Daily Functioning | IADL dependence | | | | | | |
| | Unsteady gait | | | | | | |
| | Falls | | | | | | |
| | Continence care | | | | | | |
| | ADL physical assistance | | | | | | |
| | ADL supervision/guidance | | | | | | |
| | Low levels of physical activity | | | | | | |
| Mental Wellbeing | Mild cognitive impairment | | | | | | |
| | Mood concerns | | | | | | |
| | Moderate / severe cognitive impairment | | | | | | |
| | Communication | | | | | | |
| | Behaviours | | | | | | |
| Quality of Life | Home environment safety concerns | | | | | | |
| | Loneliness | | | | | | |
| | Risk of abuse or neglect | | | | | | |
| Participation | Caregiver distress | | | | | | |
| | Informal support concerns | | | | | | |
| | Social participation | | | | | | |

instability, daily pain, cardiorespiratory symptoms, and medication concerns. Compared with other groups, Group A is between 2.5 and 9 times as likely to have symptoms of dehydration, between 1.3 and 6 times as likely to exhibit indicators of delirium, and 2 to 3 times as likely to have indicators of abuse, all rare but important events. In terms of Mental Well-Being, rates of mood disorders and cognitive impairment are among the highest in the six groups, and one quarter of Group A clients exhibited some form of behaviours. Extensive Daily Functioning needs are present, with similar IADL dependency as other groups but greater ADL dependency, with over 40% requiring physical assistance. Caregiver distress, informal support and home environment safety concerns top this group's Quality of Life and Participation needs, however over 20% report feeling lonely, with a similar number reporting distress due to declining social participation. Finally, compared with other groups, Group A is up to 9 times as likely to have symptoms of dehydration, up to 6 times as likely to exhibit indicators of delirium, and up to 3 times as likely to have indicators of abuse, all rare but important events.

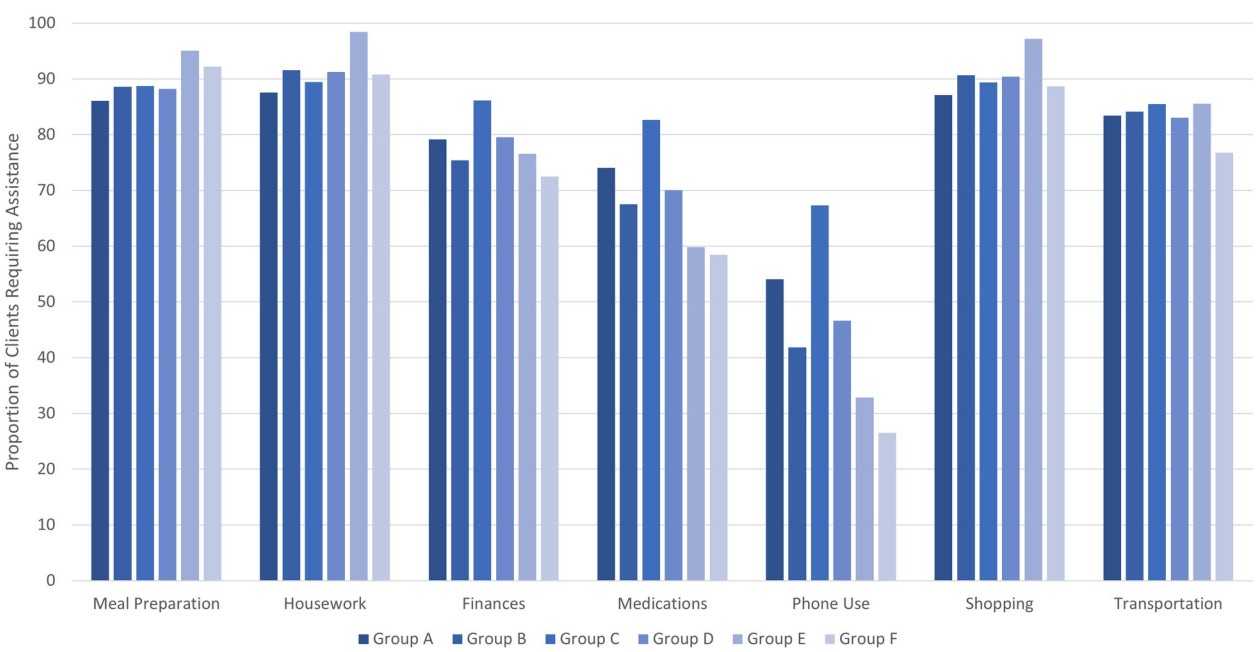

**Fig 1. Proportion of clients requiring assistance by IADL type and group.**

Needs related to Bodily Functions dominate Group B's (n = 30,793) profile with high rates of medical instability, cardiorespiratory symptoms, daily pain, and medication concerns. This group has the greatest need for medical interventions and treatments such as IV therapy, radiation treatments, dialysis etc. Notably, over half of clients in this group also live with mild cognitive impairment, potentially impacting their ability to manage medical needs. Quality of Life and Participation needs focus on home safety, loneliness, and socialization.

Clients in Group C (n = 21,611) have comparatively fewer needs related to Bodily Functions but remain at very high risk for LTC admission due to severe cognitive impairment and related effects on daily functioning and informal support. Most group members have moderate to severe cognitive impairment, with 37% experiencing communication difficulties and almost 70% exhibiting behaviours. Heavy functional needs are observed, including IADL dependance, supervision or physical help with ADLs and the greatest proportion requiring continence care. Over half of caregivers in this group report feeling distressed. Compared with other groups, members of Group C are up to 3 times as likely to have indicators of abuse.

Group D (n = 50,314) is the largest sub-group observed. While rates of care needs are not as high as other groups, Group D members still experience many needs related to Daily Functioning and Mental Well-Being at significant rates. Most group members exhibit some level of cognitive impairment, with most (60%) experiencing mild impairment. Medical instability, pain and falls continue to be an issue, while almost half (45%) live alone. Functionally, most are dependent in IADLs and over half require some form of ADL assistance.

Clients in Group E (n = 9,674) are affected by several needs related to Bodily Functions including pain, medical instability, and cardiorespiratory symptoms. Although still an infrequent need, this group has the highest rates of catheters and ostomies and one in five require wound care. In terms of Daily Functioning, almost all clients are dependent in IADLs, and most require assistance with ADLs. Notably, Participation and Quality of Life concerns of Group E are the highest across the six groups, with 85% of caregivers in this group report

feeling distressed, and almost half indicating they can no longer continue in their caregiving activities. Home environment safety and social concerns also dominate.

All clients in Group F (n = 18,850) are living with mild cognitive impairment, however the majority do not have a co-resident caregiver. Further, almost 40% are experiencing communication difficulties. This group's profile reflects greater physical independence; however, the majority still require assistance with Daily Functioning in the form of supervision or guidance with ADLs, or assistance with IADLs. Dominant needs related to Bodily Functions include pain, medical instability, and cardiorespiratory symptoms.

## Co-occurrence of life care needs

Fig 2 illustrates the proportion of clients with 'life care' needs spanning multiple Positive Health dimensions. In line with our description of the complexity of care needs for Group A above, this group has the largest proportion of clients with care needs spanning all five dimensions. However, clients in all six groups had care needs spanning multiple Positive Heath domains, emphasizing the need for a multidimensional and holistic approach to assessment, care planning and delivery in home care.

## Naming the groups to prepare for action-oriented model design

To facilitate use of the home care client profiles for authentic engagement of experts-by experience in subsequent model design steps of the mixed method study [44], and for applied use in planned practice and education initiatives, Groups A-F were given a name to represent the focus for action-oriented care model development. As the groups were segmented hierarchically, it was important that names reflected each group's unique care needs placing them at

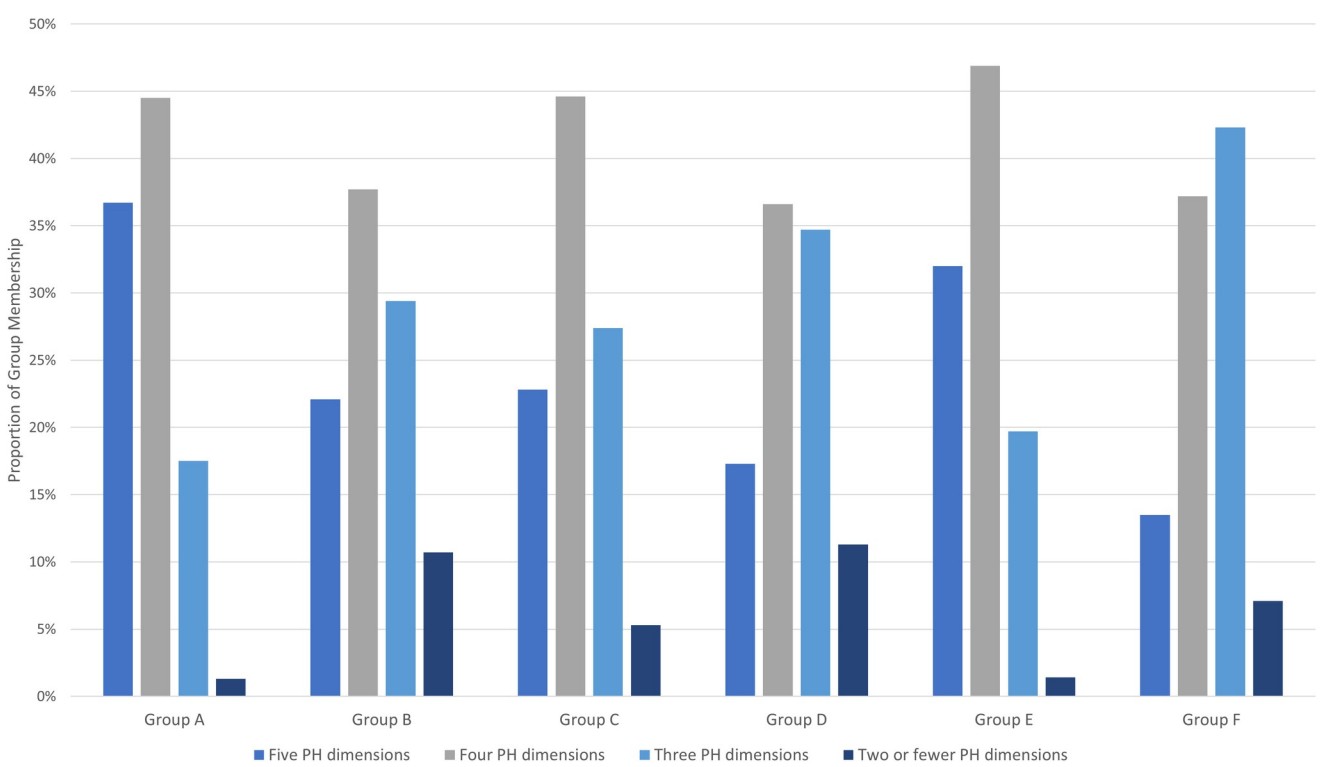

**Fig 2. Proportion of group membership with care needs across multiple positive health dimensions.**

risk for facility based LTC. As such, group names are not intended to be representative of all needs in each group. For example, high levels of caregiver distress were observed in Groups A, C and E, however in Groups A and C other care needs were observed that also placed them at risk for facility-based care admission. Group names were informed by literature on key factors influencing facility-based care admission and were chosen to highlight dominant care needs and characteristics of each group most aligned with these factors. Language used in naming was intentionally chosen to be accessible to a broad audience including aging Ontarians, caregivers and health and social care providers participating in subsequent study phases.

Group A, the **Geriatric Syndromes** group, describes a medically and socially complex, high needs group of aging Ontarians with multiple indicators of geriatric syndromes such as functional limitations, falls, depression and medication concerns including polypharmacy and related side effects. In Group B, the **Medical Complexity** group, clients have significant medical complexity that would benefit from nurse oversight and strong integration with primary care. Care needs in Group C, the **Cognitive Impairment and Behaviours** group, are dominated by cognitive and behavioural health issues and their sequelae. Given the co-occurrence of mild cognitive impairment with chronic diseases such as diabetes, care needs in Group D, the **Chronic Disease Management** group, are related to management of those conditions. The profile of Group E, the **Caregiver Distress** group, typifies characteristics associated with caregiver distress, where co-resident spousal caregivers are providing extensive daily support for functional and chronic care needs to a younger care recipient. Finally, risk for facility-based care in Group F is related to loss of resources, activities, or abilities important to meeting basic social needs, known as **Social Frailty** [64]. Most members of this group live alone and have caregivers outside their nuclear family (e.g., spouse or child).

## Discussion

In this study, a comprehensive picture of the 'life care' needs of home care clients in Ontario, Canada was generated leveraging a broadened definition of health. Six unique home care client groups were classified, their dominant care needs were examined in depth and aligned with predictors of LTC facility admission identified in the literature. The range of care needs observed, and their relative intensity across the six groups, highlights the variation in home care client needs and why a differentiated and person-centred approach to care is required to effectively meet client needs. To support the use of these profiles to design targeted home and community-based service offerings and education, some key insights regarding the needs and complexity of the home care population are offered.

Most home care models are focused on the delivery of personal support services to assist clients to complete basic self-care activities and daily tasks [19, 65]. However, in this study we observed that, across the six client groups, between 70% and 100% of study clients had one or more documented medical needs in the Bodily Functions dimension which may benefit from stronger nursing and primary care involvement, alongside the anticipated high rates of needs related to Daily Functioning. For example, daily or severe pain was observed in high rates in the population, ranging from 43% to 69% across groups, aligning with previous prevalence studies in home care [66, 67]. Uncontrolled pain has been linked with functional decline and unplanned admissions to facility-based care, contributing negatively to both individual and system outcomes [68]. Therefore, enhanced roles for nursing and allied health in home care coordination and delivery should be considered in new models to address medical needs and provide health promotion, while continuing to meet IADL and ADL needs [69].

While professional services in home care are typically reserved for task-focused post-acute care (e.g., physiotherapy following hip or knee surgery, nursing for wound care), previous

work examining effectiveness of care models integrating home and community care services and preventative health monitoring and health promotion show improved outcomes compared to usual home care [31, 70–73]. Additionally, given that across all groups, care needs related to the Daily Functioning dimension were between 96% and 100%, we see clear roles for rehabilitation therapists and therapy assistants, optimizing daily functioning and promoting physical activity to prevent frailty over the long-term [74]. Unfortunately, to date, emerging interprofessional home care models adopting a reablement or restorative care lens have been targeted at post-acute hospital-to-home clients rather than the broader home care population [75]. Study results demonstrate a broader need for these types of approaches and could be used to support the policy argument for shifting additional funding into integrated home and community care programs.

Many study clients had care needs identified in the Quality of Life and Participation dimensions of Positive Health including caregiver distress or informal care concerns, home environment safety concerns, loneliness, and social participation needs in addition to their medical, cognitive, and functional care needs. In the current task-oriented care system, social care needs aren't often integrated into care plans, despite identification through mandated assessment tools [60, 76]. Previous research demonstrates clients with co-occurring unmet functional and social care needs have twice the rate of adverse consequences such as going without eating or not taking medications as prescribed [3], highlighting the importance of addressing both in new models of care. Findings support the creation, implementation, and evaluation of an integrated, team-based model of care which assesses <u>and</u> responds to the full range of health and social care needs identified in the population [77, 78].

While many care needs were common across the six groups, others were concentrated in one or more groups, suggesting a differentiated approach to service planning is necessary across the groups and one size doesn't fit all. While individualized care plans should be generated at the client level, understanding the need for specialized skills, as well as facilitating linkages with services outside the traditional scope of home care, will be important for the development of an effective integrated home care model which serves the full breath of clients seen in home care. For example, clients in Groups A and B were more than twice as likely to experience medical instability and cardiorespiratory symptoms than the other groups, highlighting the importance of planning for primary nursing in these groups with strong linkages to primary care and specialists for effective symptom management [79]. Group C was up to 100x more likely to exhibit behaviours than the other groups, emphasizing the need for additional training in dementia care for care providers supporting these clients, as well as strong linkages to community-based services [80]. Using the grouping approach put forward in this study, organizations and health systems can better understand the proportion of their population in each care profile to identify and develop the necessary collaborations and prioritize education and training initiatives to meet life care needs.

Broadening service options alone will not be sufficient to shift the system to providing 'life care' predicated on a more holistic definition of health like Positive Health. To realize this vision education and training are needed, along with clinical and system leadership to build clinical knowledge, culture and system structures required to work in this way [77]. Research examining home care providers' knowledge and skills identified gaps in assessment and interventions for mental health and addictions, chronic disease management, and caring for clients with cognitive impairments or increased medical acuity [81–83]. Availability of high quality, accessible and targeted professional development related to key clinical issues will be required, coupled with clear expectations and time allocated for team members to upgrade skills [82]. Given the range of care needs present, it is likely the care team will need to be dynamic in membership, with some professionals being consulted for their clinical expertise as needed.

For example, given the prevalence of geriatric syndromes and the level of medical instability, cardiorespiratory symptoms and polypharmacy observed in Groups A and B, strong integration with primary care and specialized geriatric services will be necessary [35, 84, 85]. This type of care environment necessitates care team members to have a strong foundation in inter-professional collaboration and communication as well as appropriate tools to support communication within and across organizational boundaries [86, 87]. Finally, given the integral role of unregulated care providers in home and community care delivery, leveraging their care contributions in a more integrated and meaningful way will be important [88].

## Application of research findings

The home care client profiles generated through this study fill an important gap in the literature by taking a learning health systems approach and a population level perspective, providing leaders at both meso (e.g., care delivery organization) and macro (e.g., funder) levels with information necessary to design and realize new evidence-informed models of care. These data allow for the creation of a responsive home care program that responds to existing and future health needs, with the goal of improving available services and client outcomes. For example, data on dominant care needs could be mapped against provider scope to identify the health human resource mix as well as intersectoral collaborations required to adequately meet client care needs. These data also provide important details related to the provider skills and competencies necessary to meet population health needs, which can serve as the basis for workforce training and development initiatives.

Next steps for this research include examining client, caregiver, and system outcomes for the six groups over time and the translation of these client profiles into a toolkit of illustrative client vignettes to guide Phase 2 of the larger research study. The toolkit of illustrative client vignettes also has the potential to be used beyond the research study to support training and education in the provision of 'life care' more broadly across the health care system to drive evidence-informed system change and build capacity for person-centred health and social care through the lens of Positive Health.

## Strengths and limitations

This study has several strengths as well as some limitations. First, in response to calls to make better use of routinely collected clinical data to provide evidence to support system reform [83, 89, 90], this paper uses home care assessment data in a prospective way to support the design a new model of long-term home care [26]. While the profiles generated here are specific to the Ontario home care context, due to the widespread adoption of interRAI assessment tools, it is possible to replicate this work to better understand local populations. One limitation of this study is that due to the cross-sectional nature of our analyses, we are unable to determine which care needs identified through comprehensive assessment were met and which went unmet in the current service model. Future research linking assessment data with service plans and outcomes will help to identify opportunities to refine service packages to ensure needs are addressed. Further, while the interRAI Home Care assessment is comprehensive, previous work to map the assessment elements of the instrument to the six dimensions of Positive Health has noted a disproportional representation of items aligning with a biomedical model of health [37]. Therefore, we were unable to assess care needs associated with the Meaningfulness dimension in this study. In order to understand the prevalence and intensity of care needs in the Meaningfulness dimension and further examine care needs associated with the Participation dimension, complementary assessment and research focused on these dimensions is needed. Finally, due to considerations of feasibility for the larger mixed methods study

noted above, comprehensive profiles for clients in the lowest two service groups were not generated. Methods utilized in this study will be applied in future work planned to better understand the needs of populations that require less intensive and complex care, such as those living and receiving care in retirement homes or through community support services.

## Conclusions

Care needs of home care clients are diverse and extend beyond support for daily functioning. More than half of the clients in this study had at least one care need in each of the medical, functional, cognitive, and social care domains depicting a population with complex care and service needs. By conceptualizing health more broadly to include aspects of physical, mental, and emotional well-being, we can better understand those 'life care' needs and develop a new model (e.g., system approach + service offerings + clinical practices) of home and community care that mitigates known risks for facility based LTC admission and provides more options for aging Ontarians to live and manage their health at home, long-term.

## Supporting information

**S1 Table. Plain language coding rules for service groups algorithm.**
(PDF)

**S1 Checklist. RECORD statement and checklist.**
(PDF)

## Author Contributions

**Conceptualization:** Margaret E. Saari, Justine L. Giosa, Paul Holyoke, George A. Heckman, John P. Hirdes.

**Data curation:** Margaret E. Saari, George A. Heckman, John P. Hirdes.

**Formal analysis:** Margaret E. Saari, Justine L. Giosa, Paul Holyoke, George A. Heckman, John P. Hirdes.

**Funding acquisition:** Paul Holyoke.

**Investigation:** Margaret E. Saari.

**Methodology:** Margaret E. Saari, Justine L. Giosa.

**Resources:** John P. Hirdes.

**Visualization:** Margaret E. Saari.

**Writing – original draft:** Margaret E. Saari, Justine L. Giosa.

**Writing – review & editing:** Margaret E. Saari, Justine L. Giosa, Paul Holyoke, George A. Heckman, John P. Hirdes.

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
