## [Decision Letter · Decision Letter 0]

7 Nov 2023

PONE-D-23-24691Defining the dominant medical, functional, cognitive, and psychosocial care needs of community-dwelling older adults in Ontario: The case for long-term ‘life care’ at homePLOS ONE

Dear Dr. Saari,

Thank you for submitting your manuscript to PLOS ONE. After careful consideration, we feel that it has merit but does not fully meet PLOS ONE’s publication criteria as it currently stands. Therefore, we invite you to submit a revised version of the manuscript that addresses the points raised during the review process.

We look forward to receiving your revised manuscript.

Kind regards,

Anuchart Kaunnil, PhD

Academic Editor

PLOS ONE

“This study was supported by SE Health, a Canadian not-for-profit social enterprise. MS, JLG, and PH receive unrestricted salary support through their roles in the SE Research Centre. GAH receives unrestricted salary support from the Schlegel Chair in Geriatric Medicine at the Schlegel-UW Research Institute for Aging.

Reviewers' comments:

Reviewer's Responses to Questions

**Comments to the Author**

1. Is the manuscript technically sound, and do the data support the conclusions?

Reviewer #1: Yes

Reviewer #2: Yes

Reviewer #3: Yes

Reviewer #4: Partly

Reviewer #5: Yes

2. Has the statistical analysis been performed appropriately and rigorously? 

Reviewer #1: Yes

Reviewer #2: Yes

Reviewer #3: Yes

Reviewer #4: No

Reviewer #5: No

3. Have the authors made all data underlying the findings in their manuscript fully available?

Reviewer #1: Yes

Reviewer #2: Yes

Reviewer #3: Yes

Reviewer #4: Yes

Reviewer #5: Yes

4. Is the manuscript presented in an intelligible fashion and written in standard English?

Reviewer #1: Yes

Reviewer #2: Yes

Reviewer #3: Yes

Reviewer #4: Yes

Reviewer #5: Yes

5. Review Comments to the Author

Reviewer #1: Table 5 is confusing. Page 17 line 220-222 indicated that “A divergent colour gradient, ranging from blue (10th percentile) to white (50th percentile) to red (90th 221 percentile), was used to visually represent the intensity of each care need across the groups”. However, in the Table, percentage and colour used in the Table are not consistentcy. For example, Group F, Medical instability 45.2% in dark blue, Cardiorespiratory symptoms 48.7% in light blue, Pain 58.6% in light blue, ostomy care 2% in red. It would be nice if you explain more how you organized the data.

Reviewer #2: Dear authors,

Page 12 line 188: The name of tha table is not same as in the head of the table.

Page 14 line 196-201: It is not identified that the data belongs to the overall sample or each sample group.

Page 17 line 244: The word 'consideration' is bigger than other words.

One last thing, please check that your all data is follow the format of the journal.

Reviewer #3: This paper investigated the dominant of medical, functional, cognitive, and psychosocial care needs of community-based home care service. I think this paper needs minor revision to improve the rationale and gap of this study. Comments for this paper is below:

This study is a good topic that impacts on more understanding of home care clients with complex care and improve the service delivery. The background and objective of study is clear. However base on the six domains of Positive health. Why this study concerns only 4 domains? Please add this information in the introduction.

In the Method section, Ethics statement, participants’ sample size and method for select sample have clearly stated. Please describe who performed the data analysis and the method of Heat map.

In the Result section, The results are following the objective of the study and easy to understand except in table 5. A divergent color gradient is not related with the percent. It can make the reader confused. It should be adjusted to make it easier to understand and follow up on research results.

The discussion and have sufficient support evidence. However the previous evidence or guidelines for dealing with problems following the results were not discuss. The application of research findings is not clear. Conclusions are consistent with the results of the study.

The specific comment for each section is in the attachments:

Reviewer #4: During my review process, I have thoroughly examined the results presented in your work. After careful consideration, I regret to inform you that I did not find any new or novel information in this study. The interpretation and statistical analysis appear to be quite basic. Regrettably, your study lacks an explanation of the specific mixed methods used. Furthermore, a significant issue is the absence of any discussion regarding the outcomes or a service plan that could clarify on the actual needs of older adults.

Reviewer #5: Title:

-I would suggest the authors revise the title to “Profiling of medical, functional, cognitive and psychosocial care needs…” as the study focuses on these four aspects among older adults in Ontario, Canada.

Abstract

-The results are missing in the abstract. It would be good if the author could improve the abstract by highlighting the study's primary results.

Introduction

- It would be great if the author could use consistent terms throughout the manuscript to make it easier for the reader to understand the context of the study. E.g., older adults.

- Some of the abbreviations need to be introduced first before they can be used throughout the manuscript, e.g. RAI-HC, IADL, SAS etc

-In the last paragraph of the introduction, please clarify why examining the client care need based on the type and intensity of services is important. Provide the citations to support your arguments.

Objective

-As stated in the introduction, the authors highlight examining the intensity of services. The study's objective could be aligned with the problem statement as stated by the authors. I could see that the authors provided the result on the intensity of care needs in Table 5.

Methods

- Since this study can stand alone, I would recommend the authors remove the first sentence, “Detailed methods of the larger mixed-methods study have been reported elsewhere”.

-The authors may explain in the introduction the significance of the study for the larger mixed-methods study.

-Sample: is there any rationale for including the age of below 65 years old when the focus is on the older adults?

-Analysis: Provide the rationale for including only level 1 to 4 service only. And why level 5 and 6 were excluded from further analysis?

-

Results

-Since the data also included the participants below 65 years old, it is challenging to generalise the findings to the older adult population in Canada

-Table 4, please report all sub-variables and ensure the total is 100%. i.e. gender, male and female, marital status, married and etc.

- I would suggest the data in Table 5 be in chart form and arrange it in rank order for the intensity so that it would be easier for the readers to look for the trends

-Figure 1: The quality of the picture is low and blurred; the authors may need to provide a high-resolution picture.

-Dominant care needs, it would be good if the discussion could focus on the dominant care needs rather than the group. I would suggest the authors use chi-square/fisher exact test to analyze the data instead of explaining it descriptively.

-Co-occurrence of care needs; similarly, it would be good if the authors could conduct a chi-square/fisher exact test for this part.

-Naming the group: the authors may want to provide the rationale for naming the group; is it important? I suggest using the table for the group name so that it is not lengthy.

Discussion

- In general, it would be good if the authors discuss the study's findings as per objectives spanning the profile life care needs and its intensity of the need. I could see the gist of it, but it can be improved.

- It would be good if you could highlight your findings in paragraph 2 of the discussion instead of discussing only something in the literature.

-Discuss the intensity of life care needs in your discussion would be great for the audience to understand the study.

Application of research findings,

-it would be good if you could state the purpose of Phase 2 of your large study rather than stating Phase 2 only.

-Could you please provide implications of the study for practice and how it fills the knowledge gap in this area?

Limitation

-Provide further clarification as to why it could not identify the clients' unmet needs.

_ This is related to your analysis. Please clarify why analysing the lowest two service groups was not feasible.

-Lighter care population? What does that mean?

Conclusion

-To enhance the expression, put the comma after the “wellbeing”, not a put stop. By conceptualising health more broadly to include aspects of physical,

mental, and emotional well-being ”. “we…

-a good conlusiom.

General comments:

-Although some parts of the manuscript were well-written, I found it difficult to follow through because of the expressions and structure of the sentences. Thus, the manuscript will benefit from proofreading and editing services to improve the conciseness and clarity of the manuscript.

-Please check the citation for the website in the manuscript as per recommended by the journal guideline.

6. PLOS authors have the option to publish the peer review history of their article (what does this mean?). If published, this will include your full peer review and any attached files.

Reviewer #1: No

Reviewer #2: **Yes: **Pachpilai Chaiwong

Reviewer #3: No

Reviewer #4: No

Reviewer #5: **Yes: **Che Daud, A.Z.

---

## [Author Response · Author response to Decision Letter 0]

23 Jan 2024

Dear Peer Reviewers and Editorial Team,

Thank you for your thoughtful comments and feedback on our submitted manuscript entitled Defining the dominant medical, functional, cognitive, and psychosocial care needs of community-dwelling older adults in Ontario: The case for long-term ‘life care’ at home. A response to each comment appears below, with the corresponding changes tracked in the revised manuscript. Comments are labelled with both the Reviewer number and given a number based on order. For Reviewer 3, who provided both summary comments and detailed comments, we have noted where comments were from the attachment (labelled “Detailed comment”). We have presented grouped comments that had a consistent theme across multiple reviewers first, and then moved on to address individual comments. For each revision we have identified where you can look for the changes in the revised manuscript (with tracked changes). We appreciate the opportunity to revise the manuscript and believe the changes made have contributed to an improved presentation of this work. A formatted version of this response to reviewers letter has been included with our revised manuscript.

Kind regards, 

Margaret Saari RN, PhD on behalf of the author team

Related Comments Grouped by Theme 

Theme: Clarity required in the presentation of care needs 

Relevant Reviewer Comments 

Reviewer 1, Comment 1: Table 5 is confusing. Page 17 line 220-222 indicated that “A divergent colour gradient, ranging from blue (10th percentile) to white (50th percentile) to red (90th percentile), was used to visually represent the intensity of each care need across the groups”. However, in the Table, percentage and colour used in the Table are not consistentcy. For example, Group F, Medical instability 45.2% in dark blue, Cardiorespiratory symptoms 48.7% in light blue, Pain 58.6% in light blue, ostomy care 2% in red. It would be nice if you explain more how you organized the data.

Reviewer 3, Comment 3: Table 5: In the Result section, The results are following the objective of the study and easy to understand except in table 5. A divergent color gradient is not related with the percent. It can make the reader confused. It should be adjusted to make it easier to understand and follow up on research results.

Reviewer 3, Detailed comment 3: Please described the method for analysis Heat map of outcome measurement.

Reviewer 3, Detailed Comment 4: Page 17 Table 5: A divergent color gradient is not related with the percent. It can make the reader confused. It should be have statement of adjusted the table to make it easier to understand and follow up on research results.

Reviewer 5, Comment 13: I would suggest the data in Table 5 be in chart form and arrange it in rank order for the intensity so that it would be easier for the readers to look for the trends

Reviewer 5, Comment 15: Dominant care needs, it would be good if the discussion could focus on the dominant care needs rather than the group. I would suggest the authors use chi-square/fisher exact test to analyze the data instead of explaining it descriptively.

Author Response 

There was consensus across multiple reviewers that the presentation of the frequency of care needs, as well as the relative intensity and clustering of care needs in a single table (Table 5) was confusing. 

We have incorporated several suggestions from across reviewers to present this complex information in a clearer way. 

• We have added additional information with associated references to the methods to better explain the heat map visualization method (lines 213-224).

• We have chosen to split the information presented into two components, a table (Table 5) which provides detailed information on the frequency of care needs identified in the groups (along with the associated statistical tests for difference between groups) and a heat map visualization (Table 6) which allows readers to see the relative intensity and clustering of care needs across groups. 

• We have ranked the care needs within each Positive Health dimension based on their frequency in Group A in both Table 5 and 6.

• We have included results of between group testing (Chi Square with a Bonferroni correction given the number of tests conducted)

We hope that these changes convey a clearer distinction of these two independent but related concepts (i.e., intensity and grouping of needs). 

Theme: Need for additional implications / application of findings for practice

Relevant Reviewer Comments 

Reviewer 3, Comment 4: The application of research findings is not clear. 

Reviewer 3, Detailed Comment 8: Page 25 line 373: Please show how to generalize the result of this study to improve the service before explaining the future study

Reviewer 4, Comment 3: a significant issue is the absence of any discussion regarding the outcomes or a service plan that could clarify on the actual needs of older adults

Reviewer 5, Comment 22: Could you please provide implications of the study for practice and how it fills the knowledge gap in this area?

Author Response 

Several reviewers identified the need for additional information regarding the application of study findings to practice. We have added additional content to this section (Lines 490-500) which outlines how study results fill a gap in existing knowledge and can be used to support home care practice improvements.

While we appreciate requests from reviewers for discussion regarding outcomes, results of this study are at a population level and are future-focused, aiming to identify dominant care needs to support the development of a new model of integrated home and community care. Therefore, discussion about how the profiles can be used to create specific service plans or assess client outcomes in the current model of care would be out of scope.

Theme: Relationship with larger mixed-methods study

Reviewer Comments 

Reviewer 4, Comment 2: Study lacks an explanation of the specific mixed methods used

Reviewer 5, Comment 7: Since this study can stand alone, I would recommend the authors remove the first sentence, “Detailed methods of the larger mixed-methods study have been reported elsewhere”. 

Reviewer 5, Comment 8: The authors may explain in the introduction the significance of the study for the larger mixed-methods study.

Author Response 

While we appreciate the study can stand alone, we believe it is important to direct readers to the protocol of the mixed methods study to situate these analyses in the larger body of work being undertaken to understand the policy, practice and educational applications, which are based on the totality of the work. However, based on reviewer comments, we agree it is necessary important to add some clarity regarding the importance of this study for the subsequent work conducted in the larger mixed methods study. We have included this additional explanation under Study Design (Lines 118 to 124). 

Theme: Figure quality 

Reviewer Comments 

Reviewer 3, Detailed Comment 5: Page 19 Line 239: The figure 1 has not shown.

Reviewer 5, Comment 14: Figure 1: The quality of the picture is low and blurred; the authors may need to provide a high-resolution picture.

Author Response 

According to Journal guidelines (https://journals.plos.org/plosone/s/figures#loc-how-to-submit-figures-and-captions), figures are not to be placed in-line but rather submitted as separate files with figure captions placed in-line. We have revised re-submitted high-resolution TIFF files (300dpi) for both figures and used PACE to ensure alignment with formatting requirements to improve picture quality.

Individual Reviewer Comments

Reviewer Comment 

Reviewer 2, Comment 1: Page 12 line 188: The name of the table is not same as in the head of the table [Table 3].

Author Response 

We have revised the table name and description of the table in the text to better align with language used throughout the manuscript (Lines 233-238, Table 3).

Reviewer Comment 

Reviewer 2, Comment 2: Page 14 line 196-201: It is not identified that the data belongs to the overall sample or each sample group.

Author Response 

We have revised the sample description to clarify that a description of the demographics of the full study sample is provided first, followed by in-depth examination of the demographics and life care needs of six unique client groups (Lines 247-249).

Reviewer Comment 

Reviewer 2, Comment 3: Page 17 line 244: The word 'consideration' is bigger than other words.

Author Response 

This has been corrected.

Reviewer Comment 

Reviewer 2, Comment 4: One last thing, please check that your all data is follow the format of the journal.

Author Response 

Study data contained in tables are formatted according to journal guidelines obtained here: https://journals.plos.org/plosone/s/tables.

Reviewer Comment 

Reviewer 3, Comment 1: This study is a good topic that impacts on more understanding of home care clients with complex care and improve the service delivery. The background and objective of study is clear. However base on the six domains of Positive health. Why this study concerns only 4 domains? Please add this information in the introduction.

Reviewer 3, Detailed Comment 1: Page 4 Line 62: Please clarify why this study concerns only 4 [Positive Health] domains? Why the domain related environment and social were excluded. Is it more effect in 4 domains in this population or there are some barrier to analysis the social and environmental domain?

Author Response 

Thank you for identifying this inconsistency in our use of Positive Health as a theoretical underpinning for our definition of Life Care. The originally submitted version reflected some of our early thinking on how to link the Pillars for Positive Health with Life Care and interRAI assessment information. Since study conception, our research team has completed additional work mapping the items of the interRAI HC to the six domains of Positive Health which has now been published in BMC Health Services Research (Fowokan et al., 2023). 

We agree that the manuscript would be improved through closer alignment of our Life Care definition and Positive Health, while being care not to substantively change the study and analyses that were conducted and used in subsequent phases of the larger research study. Therefore, we have taken the following actions to improve alignment: 

1. We have linked the concept of Life Care with the domains of Positive Health and the interRAI Home Care comprehensive assessment instrument in the introduction (Lines 93-94)

2. We have aligned the measurement approach section and table to reflect the domains of Positive Health (Lines 233-238 and Table 3)

3. We have elaborated on environmental aspects of the Quality of Life dimension by including measures of both physical and psychological safety concerns (Table 3 and 5, Figure 1). 

4. We have reframed the presentation, labelling and discussion of results in the study using Positive Health rather than the four domains of medical, functional, cognitive and psychosocial care. (Table 5, Figure 1, Figure 3; Throughout discussion)

5. We have included in our limitations the challenges related to the underrepresentation of the PH dimensions of “meaningfulness” and “participation” in the RAI-HC data (Lines 519-530)

Reviewer Comment 

Reviewer 3, Comment 2: In the Method section, Ethics statement, participants’ sample size and method for select sample have clearly stated. Please describe who performed the data analysis and the method of Heat map.

Author Response 

It has been clarified that data analyses were conducted by the study lead author (Lines 207-208) using SAS Version 9.4

Reviewer Comment 

Reviewer 3, Detailed Comment 2: Page 8 Line 149: How is the population was grouped? Please explained the method of algorithm to select group

Author Response 

We would like to refer the reviewer to the inclusion criteria for each group which were included in Tables 1 and 2 of the original submission. We have added additional information regarding how each client is assessed against inclusion criteria in a hierarchical fashion (Lines 175-180) and we have included plain language coding rules for the service groups algorithm as a Supplementary File (Supplementary File 1) for those interested in this level of detail. 

Reviewer Comment 

Reviewer 3, Detailed Comment 6: Page 24 line 245: Please give some evidence to manage the barrier of time limited of . thrapists or interfrofessional focus on post-acute hospital

Author Response 

We have revised the paragraph discussing expanded roles for rehabilitation therapists (Lines 429-441). Evidence cited (Sims-Gould et al., 2017) is a systematic review of interventions in current home-based reablement, reactivation, rehabilitation and restoration programs. Programs reviewed by Sims-Gould focused on a post-acute population, but based on needs identified in this study, we feel therapists and therapy assistants could play an integral role on the long-stay home care team focusing on maintenance of function, prevention of frailty and quality of life. We recognize there may be operational barriers to expanded roles in the current system, however, discussion of mitigation strategies is beyond the scope of this manuscript. We have added a sentence to clarify that findings may be able to be leveraged to support policy arguments requesting additional funds for integrated care (supporting inclusion of therapy providers).

Reviewer Comment 

Reviewer 3, Detailed Comment 7: Page 24 line 353: In addition to social care, Caregiver education is also importance in psychological care and there are some results related to the caregiver. Please discuss on this topic

Author Response 

We agree that our study identifies some important results related to caregiver needs. However, focus of these analyses were on the needs of home care clients and thus, we feel discussion of caregiver needs and potential services / solutions are out of scope for this paper. Future papers that will result from this project will address the centrality of the caregiver role in the provision of home care and will feature caregivers in engagement activities as well as model design principles and outputs.

Reviewer Comment 

Reviewer 4, Comment 1: During my review process, I have thoroughly examined the results presented in your work. After careful consideration, I regret to inform you that I did not find any new or novel information in this study. The interpretation and statistical analysis appear to be quite basic.

Author Response 

Thank you for taking the time to review our submitted manuscript. We hope that the revisions completed based on the other reviewers’ feedback may highlight the unique contributions this study makes to the evidence. By leveraging Positive Health to understand the types of needs in the home care population, we start to build the argument for shifting the system towards a more integrated approach which considers a person’s overall well-being and provides more options for living and managing health at home. 

Reviewer Comment 

Reviewer 5, Comment 1: I would suggest the authors revise the title to “Profiling of medical, functional, cognitive and psychosocial care needs…” as the study focuses on these four aspects among older adults in Ontario, Canada.

Author Response 

We have revised the title to be more inclusive of all persons assessed for home care services regardless of age. We feel this better reflects the goals and population of the manuscript.

Reviewer Comment 

Reviewer 5, Comment 2: The results are missing in the abstract. It would be good if the author could improve the abstract by highlighting the study's primary results.

Author Response 

We have revised the abstract to include the study’s primary results.

Reviewer Comments 

Reviewer 5, Comment 3: It would be great if the author could use consistent terms throughout the manuscript to make it easier for the reader to understand the context of the study. E.g., older adults.

Reviewer 5, Comment 9: Sample: is there any rationale for including the age of below 65 years old when the focus is on the older adults?

Reviewer 5, Comment 11: Since the data also included the participants below 65 years old, it is challenging to generalize the findings to the older adult population in Canada

Author Response 

Thank you for your comments. As noted above, we have revised the language used in the title and paper to be more inclusive of all persons assessed for home care services, regardless of age. Because we are focusing on long-stay home care clients, our study sample is predominantly, but not exclusively over 65. In the study sample, a total of 19,936 (12.3%) cases were under the age of 65, with the proportion varying with group membership. In this work, and in the larger mixed methods study, we have taken an aging-focused approach to model development, recognizing that there is no specific age cut-off that indicates complexity. For example, a person living with intellectual and developmental disabilities served in a home care program may be considered to have “complex” needs with high levels of frailty at a much younger age. We have gone through the manuscript to replace the term “older adult” with “adult” or “aging Ontarians” to reflect this.

Reviewer Comment 

Reviewer 5, Comment 4: Some of the abbreviations need to be introduced first before they can be used throughout the manuscript, e.g. RAI-HC, IADL, SAS etc

Author Response 

We have reviewed the use of abbreviations in the manuscript and ensured terms are introduced prior to their use throughout the manuscript.

Reviewer Comment 

Reviewer 5, Comment 5: In the last paragraph of the introduction, please clarify why examining the client care need based on the type and intensity of services is important. Provide the citations to support your arguments.

Author Response 

We have updated the paragraph and included additional citations to support our position (Lines 97-106). Grouping clients based on the types and intensity of their care needs, rather than by diagnosis, location or even care approach takes into account the complexity added to the care situation through the convergence of medical, functional, social, and environmental factors.

Reviewer Comment 

Reviewer 5, Comment 6: As stated in the introduction, the authors highlight examining the intensity of services. The study's objective could be aligned with the problem statement as stated by the authors. I could see that the authors provided the result on the intensity of care needs in Table 5.

Author Response 

We have revised the objective and primary research question (Lines 108-114) to reflect our initial hypothesis that there were distinct groups within the home care population and that these groups would have different dominant care needs that align to known predictors of admission to facility-based long-term care. 

Reviewer Comment 

Reviewer 5, Comment 10: Analysis: Provide the rationale for including only level 1 to 4 service only. And why level 5 and 6 were excluded from further analysis?

Author Response 

We have added rationale for focusing on Service Levels 1 through 4 and excluding Service Levels 5 and 6 in the manuscript (Lines 195-201). 

Reviewer Comment 

Reviewer 5, Comment 12: Table 4, please report all sub-variables and ensure the total is 100%. i.e. gender, male and female, marital status, married and etc.

Author Response 

Thank you for your suggestion. However, given the complexity of the table, for dichotomized variables only one category is shown to reduce redundancies. This is consistent with other recently published PLOS ONE papers using interRAI data (Wang et al., 2023 )

Reviewer Comment 

Reviewer 5, Comment 16: Co-occurrence of care needs; similarly, it would be good if the authors could conduct a chi square/fisher exact test for this part.

Author Response 

Upon reflection, we have chosen to include information about the co-occurrence of care needs as a figure (Fig. 2) to better visualize the differences across groups, aiding in interpretability. 

Reviewer Comment 

Reviewer 5, Comment 17: Naming the group: the authors may want to provide the rationale for naming the group; is it important? I suggest using the table for the group name so that it is not lengthy.

Author Response 

Thank you for your suggestion to potentially reduce the manuscript word count. We feel that naming of the groups is critical for applied use of this data and have added a sentence to reflect that (Lines 372-374). We did attempt to move this content into table format; however, given the need to explain the group naming, we were unable to reduce much of the content. Therefore, we have decided to keep this sub-section as prose, rather than change the format.

Reviewer Comment 

Reviewer 5, Comment 18: In general, it would be good if the authors discuss the study's findings as per objectives spanning the profile life care needs and its intensity of the need. I could see the gist of it, but it can be improved.

Reviewer 5, Comment 19: It would be good if you could highlight your findings in paragraph 2 of the discussion instead of discussing only something in the literature.

Reviewer 5, Comment 20: Discuss the intensity of life care needs in your discussion would be great for the audience to understand the study.

Author Response 

We have revised the discussion to situate study findings more directly within the literature, as it relates to life care needs. We first highlight findings related to the broader positive health dimensions and their implications for the design of home and community care programs and associated education initiatives. We then added an additional paragraph discussing specific care needs which have greater variation across groups, with some potential service considerations. 

Reviewer Comment 

Reviewer 5, Comment 21: it would be good if you could state the purpose of Phase 2 of your large study rather than stating Phase 2 only.

Author Response 

We have made this change. (Line 503)

Reviewer Comment 

Reviewer 5, Comment 23: Provide further clarification as to why it could not identify the clients' unmet needs. 

Author Response 

We have added additional details regarding the cross-sectional nature of the analyses and how future longitudinal studies are required to link assessment data with service plans and outcomes to identify which needs are met and which went unmet in the current service models. (Lines 515-519) 

Reviewer Comment 

Reviewer 5, Comment 24: This is related to your analysis. Please clarify why analysing the lowest two service groups was not feasible.

Author Response 

More detailed information regarding the decision to restrict the sample to the top four service levels has been included in the Analysis section. (Lines 195-201) 

Reviewer Comment 

Reviewer 5, Comment 25: Lighter care population? What does that mean?

Author Response 

We have revised this sentence for clarity. (Lines 533-535) 

Reviewer Comment 

Reviewer 5, Comment 26: To enhance the expression, put the comma after the “wellbeing”, not a put stop. a good conlusiom.

Author Response 

We have revised this sentence. (Lines 540-545) 

Reviewer Comment 

Reviewer 5, Comment 27: Although some parts of the manuscript were well-written, I found it difficult to follow through because of the expressions and structure of the sentences. Thus, the manuscript will benefit from proofreading and editing services to improve the conciseness and clarity of the manuscript.

Author Response 

We have made edits throughout the manuscript to improve clarity and conciseness. 

Reviewer Comment 

Reviewer 5, Comment 28: Please check the citation for the website in the manuscript as per recommended by the journal guideline.

Author Response 

We have corrected the citations for the two websites cited in the manuscript (reference 47 and 48).

---

## [Decision Letter · Decision Letter 1]

29 Feb 2024

Profiling the medical, functional, cognitive, and psychosocial care needs of adults assessed for home care in Ontario, Canada: The case for long-term ‘life care’ at home

PONE-D-23-24691R1

Dear Dr. Saari,

We’re pleased to inform you that your manuscript has been judged scientifically suitable for publication and will be formally accepted for publication once it meets all outstanding technical requirements.

Kind regards,

Anuchart Kaunnil, PhD

Academic Editor

PLOS ONE

Additional Editor Comments (optional):

Reviewers' comments:

Reviewer's Responses to Questions

**Comments to the Author**

1. If the authors have adequately addressed your comments raised in a previous round of review and you feel that this manuscript is now acceptable for publication, you may indicate that here to bypass the “Comments to the Author” section, enter your conflict of interest statement in the “Confidential to Editor” section, and submit your "Accept" recommendation.

Reviewer #2: All comments have been addressed

Reviewer #3: All comments have been addressed

2. Is the manuscript technically sound, and do the data support the conclusions?

Reviewer #2: Yes

Reviewer #3: Yes

3. Has the statistical analysis been performed appropriately and rigorously? 

Reviewer #2: Yes

Reviewer #3: Yes

4. Have the authors made all data underlying the findings in their manuscript fully available?

Reviewer #2: Yes

Reviewer #3: Yes

5. Is the manuscript presented in an intelligible fashion and written in standard English?

Reviewer #2: Yes

Reviewer #3: Yes

6. Review Comments to the Author

Reviewer #2: (No Response)

Reviewer #3: This research can be published in this journal. This paper investigated the dominant medical, functional, cognitive, and psychosocial care needs of community-based home care service. This study is a good topic that impacts on more understanding of home care clients with complex care and improve the service delivery. The rationale of the study is clearer. The result in Table 5 can be easily understood after adjustment. The authors give more information in the application of research that reviewer comment.

7. PLOS authors have the option to publish the peer review history of their article (what does this mean?). If published, this will include your full peer review and any attached files.

Reviewer #2: No

Reviewer #3: No

---

## [Editor Report · Acceptance letter]

21 Mar 2024

PONE-D-23-24691R1 

PLOS ONE

Dear Dr. Saari, 

I'm pleased to inform you that your manuscript has been deemed suitable for publication in PLOS ONE. Congratulations! Your manuscript is now being handed over to our production team.

Kind regards, 

on behalf of

Dr. Anuchart Kaunnil 

Academic Editor

PLOS ONE